# Gold Nanoparticles Using *Ecklonia stolonifera* Protect Human Dermal Fibroblasts from UVA-Induced Senescence through Inhibiting MMP-1 and MMP-3

**DOI:** 10.3390/md18090433

**Published:** 2020-08-19

**Authors:** Eun-Sook Jun, Yeong Jin Kim, Hyung-Hoi Kim, Sun Young Park

**Affiliations:** 1Biomedical Research Institute, Pusan National University Hospital, Busan 49241, Korea; mussojun@hanmail.net; 2Department of Laboratory Medicine, Pusan National University Hospital, Busan 49241, Korea; yjkim86@nate.com; 3Bio-IT Fusion Technology Research Institute, Pusan National University, Busan 46241, Korea

**Keywords:** gold nanoparticle, *Ecklonia stolonifera*, UVA, human dermal fibroblasts

## Abstract

The effect of gold nanoparticles (GNPs) synthesized in marine algae has been described in the context of skin, where they have shown potential benefit. *Ecklonia stolonifera* (ES) is a brown algae that belongs to the Laminariaceae family, and is widely used as a component of food and medicine due to its biological activities. However, the role of GNPs underlying cellular senescence in the protection of *Ecklonia stolonifera* gold nanoparticles (ES-GNPs) against UVA irradiation is less well known. Here, we investigate the antisenescence effect of ES-GNPs and the underlying mechanism in UVA-irradiated human dermal fibroblasts (HDFs). The DPPH and ABTS radical scavenging activity of ES extracts was analyzed. These analyses showed that ES extract has potent antioxidant properties. The facile and optimum synthesis of ES-GNPs was established using UV-vis spectra. The surface morphology and crystallinity of ES-GNPs were demonstrated using high resolution transmission electron microscopy (HR-TEM), energy dispersive spectroscopy (EDS), X-ray diffraction (XRD), and Fourier-transform infrared spectroscopy (FT-IR). ES-GNPs presented excellent photocatalytic activity, as shown by the photo-degradation of methylene blue and rhodamine B. A cellular senescence model was established by irradiating HDFs with UVA. UVA-irradiated HDFs exhibited increased expression of senescence-associated β-galactosidase (SA-β-galactosidase). However, pretreatment with ES-GNPs resulted in reduced SA-β-galactosidase activity in UVA-irradiated HDFs. Intracellular ROS levels and G1 arrest in UVA-irradiated HDFs were checked against the background of ES-GNP treatment to investigate the antisenescence effects of ES-GNPs. The results showed that ES-GNPs significantly inhibit UVA-induced ROS levels and G1 arrest. Importantly, ES-GNPs significantly downregulated the transcription and translation of MMP (matrix metalloproteinases)-1/-3, which regulate cellular senescence in UVA-irradiated HDFs. These findings indicate that our optimal ES-GNPs exerted an antisenescence effect on UVA-irradiated HDFs by inhibiting MMP-1/-3 expression. Collectively, we posit that ES-GNPs may potentially be used to treat photoaging of the skin.

## 1. Introduction

Skin photoaging happens as a result of exposure to ultraviolet (UV) light. Only 5% of solar radiation reaches the Earth’s surface, comprising wavelengths in the ranges of 315–400 nm (ultraviolet A (UVA)), 280–315 nm (UVB), 100‒280 nm (UVC). Specifically, UVA accounts for approximately 96.5% of the daily UV irradiation [1,2]. UVA irradiation can penetrate the epidermal and dermal layers of the skin, and contribute to oxidative stress, premature skin aging, and photo-carcinogenesis. Sustained exposure to UVA affects cell growth and survival, and induces DNA damage, the production of reactive oxygen species (ROS), and senescence-associated β-galactosidase (SA-β-galactosidase) activity [3,4]. Repeated exposure of human dermal fibroblasts (HDFs) to UVA irradiation has been established as a cellular senescence model to investigate certain characteristics of epidermal and dermal skin photoaging. When HDFs are exposed to UVA irradiation, there is an increase in the production of ROS, which leads to apoptosis, upregulation of matrix metalloproteinases (MMPs) expression, and the induction of senescence. UVA irradiation has also been reported to directly or indirectly lead to the release of inflammatory mediators, as well as pro-inflammatory cytokines. All these UVA-induced factors are responsible for macro- and micro- structural deterioration of human skin [5,6,7].

In skin, UVA irradiation can lead to the induction of collagen fiber disorganization and injury, substantial deposition of abnormal elastic fibers, and upregulation of glycosaminoglycans. Prolonged exposure to UVA results in the degradation of collagen and elastin, and reduction in the synthesis of collagen [8]. MMPs are zinc-containing endopeptidases that are able to digest various components of the dermal extracellular matrix (ECM), such as fibrillary collagen, elastin, laminin, proteoglycans, and fibronectin. They can be classified into five groups based on structural similarity, as well as substrate specificity: collagenases (MMP-1, MMP-8, and MMP-13), gelatinases (MMP-2 and MMP-9), stromelysins (MMP-3, MMP-10, and MMP-11), matrilysins (MMP-7, and MMP-26), and membrane-type-MMPs (MMP-14, MMP-15, and MMP-16). These proteins, and particularly MMP-1 and MMP-3, are thought to play a critical role in UVA-induced skin photoaging [2,9,10]. MMP-1 and MMP-3 are mainly secreted by HDFs, and degrade fibrillary collagen and type IV collagen. Studies on skin photoaging have shown that UVA promotes the degradation of ECM, as evidenced by a reduction in soluble collagen in the dermis that is associated with the upregulation of MMP-1 and MMP-3 [1,9].

Secondary metabolites derived from marine algae have been identified as a novel class of agents that can be used as cosmetic ingredients, as they have topical antibacterial, antiwinkle, anti-inflammatory and antimelanogenic properties [11]. The secondary metabolites of marine algae play an important role in the green synthesis of gold nanoparticles (GNPs) containing active substances with antioxidant properties. The green synthesis of GNPs could also be mediated by marine algae extracts containing molecules that serve as reducing, stabilizing, and capping agents [12,13,14]. GNPs have been used to treat a number of diseases by virtue of their biocompatibility and unique properties, such as conductivity, optical catalytic activity, and better structural characteristics than their bulk counterparts. GNP-based marine algae have been investigated thoroughly, and it has been experimentally documented that they possess medicinal properties, as well as various biological effects, such as antioxidative, antibacterial, anti-inflammatory, and antitumor activities [15,16,17,18]. *Ecklonia stolonifera* is a brown alga of the Laminariaceae family that is mainly distributed in the sea forests off the coasts of Far Eastern countries, such as Korea, China, and Japan [19]. The pharmacological effects of *Ecklonia stolonifera* can be attributed to the plant’s high levels of phlorotannins, phenolic compounds, terpenoids, steroids, and vitamins, all of which are associated with a range of effects, such as antioxidative, antimicrobial, anti-inflammatory, antiallergic, hepatoprotective, neuroprotective, antidermal-aging activities [20,21,22,23,24,25,26]. The present study demonstrated that the *Ecklonia stolonifera* extract is more effective than other conventional methods for the efficient synthesis of GNPs due to the presence of a large number of secondary metabolites which are required for the reduction, capsulation, and stabilization of GNPs. To the best of our knowledge, this study is the first to explore the potential antiphotoaging effects of ES-GNPs in UVA irradiated HDFs.

## 2. Results

### 2.1. Optimization of ES-GNPs Using the ES Reductant

In this present study, GNPs were synthesized using the ES extract as the reductant. To determine the potential effect of the ES extract on the reduction, capsulation and stabilization of GNPs, the extracts were investigated for radical scavenging (ABTS (2,2′-Azino-bis(3-ethylbenzothiazoline-6-sulphonic acid) and DPPH (2,2-diphenyl-1-picrylhydrazyl)) activity. The results of the radical ABTS and DPPH assays revealed that the radical scavenging activity was markedly higher after treatment with the ES extract (Figure 1A). We assessed the potential of the ES extract to reduce a gold (III) chloride solution that resulted in the formation of GNPs. The addition of the ES extract to the gold (III) chloride solution at room temperature resulted in the formation of ES-GNPs, which was evident from a change in color (light-yellow to ruby red). Freshly prepared ES-GNPs in the ES extract exhibited a strong absorption at 543 nm in the UV region (Figure 1B). The optimized parameters for ES-GNPs were taken into consideration for synthesizing the ES-GNPs in a simple manner. Reaction parameters for optimizing the gold nanoparticles, such as ES extract concentration, gold(III) chloride solution concentration, and reaction times, were taken into consideration to purify the GNPs in a quick and simple manner. Therefore, a concentration of 1 mg/mL ES extracts (Figure 1C), 1 µM gold (III) chloride (Figrue 1D) and a reaction time of 20 min were chosen (Figure 1E). The resulting ES-GNPs were examined using Dynamic Light Scattering (DLS) to study their hydrodynamic size distribution, zeta potential, and polydispersity index (PDI). The hydrodynamic size distribution, zeta potential and the PDI value of the ES-GNPs were found to be 49.5 ± 1.0 nm,−25.23 ± 1.1 mV, and 0.287 ± 0.001, respectively (Figure 1F,G).

### 2.2. ES-GNPs Characterization by HR-TEM/EDS

In this study the surface morphology of ES-GNPs was investigated via the HR-TEM/EDS analysis. Typical HR-TEM micrographs displayed many spherical particles, and the ES-GNPs were found to have an average diameter of 27.9 ± 4.3 nm with hexagonal-shaped morphology (Figure 2A,B). The SAED Pattern showed similar diffraction patterns which confirm the nanocrystalline nature of the ES-GNPs (Figure 2C). The FFT pattern of ES-GNPs also indicated a face-centered cubic crystal structure, which exhibited bright circular spots (lattice planes of Bragg’s reflection (111), (200), (220), and (311) planes) (Figure 2D). Figure 2E shows the representative red-particle image of the GNPs and the corresponding map for the gold (Au) atoms. The distribution of the Au atoms in the ES-GNPs was studied using High-angle annular dark-field scanning transmission electron microscopy (Figure 2F). The EDS technique represents a precise tool for analyzing the elemental composition of ES-GNPs. The results showed peak indexing of Au at 0.17 keV, 2.12 keV, and 9.81 keV; this corresponded to the results of the surface plasmon resonance analysis of the ES-GNPs that can be attributed to the presence of gold (Figure 2G). These observations suggest that it is possible to synthesize high quality ES-GNPs using our simple method.

### 2.3. Physicochemical Characterization of ES-GNPs

To examine the crystallization pattern of the gold atoms in the ES-GNPs, XRD measurements were recorded, as represented in Figure 3A. The main peaks were obtained at (111), (200), (220), and (311) corresponding to the Bragg’s reflections, with 2θ values (30–80°) of 37.19°, 43.35°, 66.82°, and 78.51°, respectively. The peak of the XRD spectrum confirmed that the highly purified GNPs were composed of crystalline gold. The “green” synthesis of GNPs involves biocompatible materials that can act as functionalizing ligands under physiological conditions, thereby aiding in the synthesis of GNPs that are more appropriate for biomedical applications [27]. ES extracts also provide possible functional groups that can attach to the surface of the ES-GNPs and form a cap, leading to efficient stabilization. The FT-IR spectra of the ES extracts and the corresponding ES-GNPs indicated extensive similarities. In particular, the prominent peak in ES—that corresponded to that in ES-GNPs—at 3401.81 cm^−1^, was related to the O–H stretch and H bonds. The peaks at 2928.15 cm^−1^ were attributed to the C–H stretch. Additionally, the peaks at 1262.45 cm^−1^ could be attributed to the C=O stretch of aromatic compounds (Figure 3B). These characteristic peaks of ES-GNPs corresponded with those in ES, indicating that the ES was successfully intercalated into the ES-GNPs. On the basis of these observations, and because of the many functional groups in the ES extract due to the presence of secondary metabolites such as phlorotannins, phenolic compounds, terpenoids, steroids, and vitamins, it was shown that our procedure could provide active sites for improving biocompatibility. The reduction of methylene blue and rhodamine B was studied using sodium borohydride in the presence of ES-GNPs, and monitored by a UV-Vis spectrophotometer. Absorption peaks corresponding to those of pure methylene blue and rhodamine B were observed at 665 nm and 555 nm, respectively. In the presence of ES-GNPs, the intensity of the deep blue (methylene blue) and pink-red (rhodamine B) colors gradually decreased and finally disappeared. In Figure 3C,D, the UV-vis spectra revealed that the peak intensity completely disappeared within 8 min of addition of ES-GNPs. Based on these data, it can be seen that we have persuasively shown that the ES extract is an important parameter for regulating the synthesis and influencing the properties of ES-GNPs.

### 2.4. ES-GNPs Ameliorate SA-β-Galactosidase Activity in UVA-Irradiated Human Dermal Fibroblasts

Before evaluating the potential antiphotoaging properties of ES-GNPs, cell viability was investigated using the Cell Count Kit 8 (CCK-8) assay. The results showed that the ES extract and ES-GNPs (up to 200 µg/mL) did not affect cell viability (Figure 4A,B). Therefore, in all the subsequent experiments, the cells were treated with ES extract or ES-GNPs, at doses of 100 µg/mL. The goal of our experiment was to examine the effect of ES-GNPs on the cellular senescence model that involved the induction of senescence by UVA irradiation of HDFs. As senescence-associated β-galactosidase (SA-β-galactosidase) is an important indicator of cellular senescence in models of skin photoaging, the activity of SA-β-galactosidase in the HDFs was determined. The results indicated that in comparison with the control group, the SA-β-galactosidase activity was markedly higher after UVA irradiation. However, pre-exposure with ES-GNPs remarkably reduced the SA-β-galactosidase activity (Figure 4C). We further confirmed the quantitative cellular senescence assay using flow cytometry, and found that senescence is elevated after UVA irradiation. The SA-β-galactosidase staining levels were remarkably higher in the UVA-irradiated group than in the control group. However, ES-GNPs could inhibit the increase observed in SA-β-galactosidase staining levels after UVA irradiation, indicating that ES-GNPs alleviate cellular senescence in UVA-irradiated HDFs (Figure 4D). Importantly, ES-GNPs inhibited the SA-β-galactosidase activity in HDFs to a greater extent against the background of UVA irradiation than the ES extracts. ES-GNPs also seemed to show more anti-SA-β-galactosidase activity than Cit-GNPs. These results suggest that ES-GNPs alleviate senescence in HDFs by inhibiting SA-β-galactosidase activity.

### 2.5. ES-GNPs Inhibit ROS Production and Lysosome Content in UVA-Irradiated Human Dermal Fibroblasts

Cellular senescence is characterized by the oxidative stress that results from the accumulation of ROS and lysosome content [28]. We conducted a CM-H_2_DCFDA staining assay using flow cytometry to assess whether ES-GNPs could effectively inhibit the intracellular ROS production in UVA-irradiated HDFs. As shown in Figure 5A, UVA-irradiated HDFs exhibited significantly increased ROS production levels compared to the control group; however, pretreatment with ES-GNPs (100 µg/mL) inhibited the intracellular ROS production in UVA-irradiated HDFs. To further investigate the antisenescence effect of ES-GNPs, the reduced lysosome content of ES-GNPs was examined in UVA-irradiated HDFs by the LysoTracker Green DND-26 detection. Figure 5B shows that irradiation with UVA markedly increased the lysosome contents in HDFs, and that ES-GNPs inhibited the lysosome contents in UVA-irradiated HDFs. The results also show that ES-GNPs significantly decreased the levels of intracellular ROS and lysosome content, compared to the ES extract at an equivalent concentration in UVA-irradiated HDFs. Thus, our data indicate that ES-GNPs suppress UVA irradiation-induced senescence in HDFs by inhibiting ROS production and lysosome content.

### 2.6. ES-GNPs Inhibit G1 Arrest and Senescence-Related Proteins in UVA-Irradiated Human Dermal Fibroblasts

Cellular senescence is defined as an irreversible cell cycle arrest, wherein cells undergo changes at the chromatin and transcript levels. The effect on the G1-arrest was confirmed using the cell cycle kit by Flow cytometry. We observed an increase in the G1-arrest in HDFs irradiated with UVA; this could be inhibited by pre-exposing cells to ES-GNPs (Figure 6A). According to the results above, we suggest that ES-GNPs alleviate HDF senescence. In order to further verify our hypothesis, we used western blot to detect senescence related proteins such as p16, p21, p-p53 and p53 in UVA-irradiated HDFs. As shown, UVA irradiation can promote the expression of p16, p21 and p-p53. However, ES-GNPs significantly decreased the expression of p16, p21 and p-p53 (Figure 6B). These results demonstrate that ES-GNPs can suppress the G1-arrest via the downregulation of p16, p21 and p-p53 expression in UVA-irradiated HDFs.

### 2.7. ES-GNPs Downregulate MMP−1/−3 mRNA, Protein Expression and Secretion in UVA-Irradiated Human Dermal Fibroblasts

HDFs are the main cells found in skin, and are responsible for the synthesis and degradation of epidermal ECM. During the progression of skin-photoaging, there is an induction in oxidative stress in the HDFs that results in ECM changes, leading to the degradation of the dermal skin layer. MMP-1/-3 play a pivotal role in skin-photoaging induced by UVA irradiation [29]. Moreover, the UVA irradiation-induced senescence in HDFs is associated with the degradation of the epidermal ECM. Taken together, these changes in UVA irradiated HDFs contribute to the initiation and progression of skin-photoaging. UVA irradiated HDFs exhibit upregulation of MMP-1/-3, which then upregulate cellular senescence in UVA-irradiated HDFs [4]. To further investigate the mechanisms of the antisenescence effect of ES-GNPs via MMP-1/-3, we studied the effects of the ES-GNPs on the transcription and translation of MMP-1/-3 in the UVA-irradiated group. Thereafter, we analyzed the mRNA level of MMP-1/-3 in UVA irradiated HDFs. Real time PCR revealed that ES-GNPs decreased the mRNA level of MMP-1/-3 in HDFs irradiated with UVA (Figure 7A,B). Furthermore, we examined whether ES-GNPs could suppress the protein expression of MMP-1/-3 in UVA irradiated HDFs. As shown in Figure 7C,D, ES-GNPs abolished UVA-induced expression of MMP-1/-3 proteins. Furthermore, pretreatment with ES-GNPs significantly downregulated the expression of MMP-1/-3 at the mRNA and protein levels relative to that in the ES extract treatment. Figure 7E,F shows that the secretion of MMP-1/-3 was increased by UVA irradiation, and that ES-GNPs reduced the secretion levels of MMP-1/-3 in UVA irradiated HDFs. In brief, ES-GNP resisted UVA-irradiated HDF senescence, and these effects are based on the downregulation of the expression and secretion of MMP-1/-3.

## 3. Discussion

UV irradiation-induced skin aging is a heavy extrinsic form of aging resulting in the formation of winkles and a reduction in the levels of collagen and elastin in human skin. UVA irradiation causes DNA damage, degradation of collagen fibers, lipid oxidation and skin aging by generating intracellular ROS [2]. Presently used antiphotoaging candidates suffer from several shortcomings such as poor water solubility, biodegradability and bioavailability [13,14]. Nanomaterial therapy is one of the promising approaches to combat the aforementioned limitations associated with most antiphotoaging candidates. Photo-aging is prevented or treated effectively by nanomaterial therapy, whereby brown alga extracts are employed. *Ecklonia stolonifera* has several biological actions including antioxidant, anti-inflammatory and antidermal-aging activities. Due to the potential of brown alga and GNPs, our study focused on the design of *Ecklonia stolonifera*-based GNPs. The particle size analysis of ES-GNPs using DLS showed average particle diameters of 49.5 ± 1.0 nm, while the PDI and zeta potential were 0.287 ± 0.001 and −25.23 ± 1.1 mV, respectively, demonstrating good dispersion. A particle size analysis of the ES-GNPs showed dimensions of 27.9 ± 4.3 nm with spherical shape morphologies, as confirmed by HR-TEM. The physicochemical properties of the ES-GNPs were successfully studied by DLS, HR-TEM, EDS, XRD and FT-IR. The EDS, XRD and FT-IR spectra exhibited expected functional groups, confirming the successful reduction, stabilization and capsulation of *Ecklonia stolonifera* extracts onto GNPs.

UVA is skin photoaging inducer causing significant cellular senescence and ECM damage [9]. Hence, there is a need to develop interventions to protect against its effects. We conducted the present study using ES-GNPs to examine cell viability, SA-β-galactosidase activity, ROS production, lysosome content, the expression of senescence related proteins, expression and secretion of MMP-1/-3 UVA irradiated HDFs. The HDF cytotoxicity of the ES extract and ES-GNPs was first evaluated using a CCK-8 assay; ES extract and ES-GNPs demonstrated cytotoxicity in a concentration-dependent way. The most common cellular senescence was characterized by SA-β-galactosidase activation. A hypothetical hydrolase, e.g., SA-β-galactosidase, is commonly used as an indirect essential maker of senescent cells. There is evidence showing that the HDF senescence caused by UVA irradiation could cause the increase of SA-β-galactosidase activity [4]. In our study, UVA irradiation caused an increase in SA-β-galactosidase activity in HDF, while intervention with ES-GNPs noticeably diminished the impact thereof.

It is generally accepted that cellular senescence can accumulate intracellular ROS production. Furthermore, a study demonstrated that a high lysosome content may play a significant role in the process of the HDF senescence, which is stimulated by UVA irradiation [3]. Our results were in accordance with previous studies, as we found essential mediators for the promotion of UVA-irradiated HDF senescence, which was also evident by the ROS production and lysosome contents. This increase was further attenuated significantly by ES-GNPs. Therefore, the inhibition of ROS production and lysosome contents was due to the ability of ES-GNPs to reduce HDF senescence by stimulating UVA irradiation. Senescence related proteins including p16, p21 and p-p53 have also been proposed to play a critical role in UVA-irradiated HDF senescence [4]. In our study, we found that treatment with ES-GNPs inhibited the expression level of p16, p21 and p-p53, indicating the dependence of p16, p21 and p-p53 in ES-GNP-initiated anticellular senescence effects.

UVA irradiation not only affects the HDF functions, but also regulates the collagen microenvironment which, in turn, promotes MMPs expression. Thus, this alteration is a prominent feature of UVA-photoaged HDF. In particular, MMP-1/-3 is frequently expressed in HDF and plays an important role in HDF senescence. MMP-1/-3 inhibitors have been shown to improve anticellular senescence effects and decrease the SA-β-galactosidase activity and expression levels of senescence related proteins [9,10]. Therefore, crosstalk between the UVA irradiated HDF senescence and MMP-1/-3 needs further investigation. In the present study, ES-GNPs were shown to hinder the expression and secretion of MMP-1/-3 that perform vital roles in the development of UVA irradiated HDF senescence.

## 4. Materials and Methods

### 4.1. Reagents

In this study, 2,2′-Azino-bis (3-ethylbenzothiazoline-6-sulphonic acid) (ABTS), 3-(4,5-dimethythiazol-2-yl)-2,5 diphenyltetrazoliumbromide (MTT), chloroauric acid (HAuCl4·3H2O), dimethyl sulfoxide (DMSO), 2,2-diphenyl-1-picrylhydrazyl (DPPH), and other chemicals were obtained from Sigma-Aldrich (St. Louis, MO, USA). All chemicals were of analytical grade.

### 4.2. Free Radical Scavenging Assay

The DPPH radical scavenging activity of ES extracts was determined by following a previously published method [30], with slight modifications. ES extracts were mixed with DPPH solution (60 µM) in 24 well microplates. The samples were shaken vigorously and then incubated at 25 °C for 2 h in the dark; then, the optical density was measured at 510 nm on a FLUOstar Omega Plate Reader (BMG Labtech, Ortenberg, Germany). The ABTS assay was performed to determine the radical scavenging activity of ES extracts in accordance with a previously published method [31], with slight modifications. ES extracts were mixed with ABTS solution (7 mM) and potassium persulfate (2.6 mM) and then incubated in the dark at 25 °C for 30 min. The absorbance was quantified at 734 nm on a spectrophotometer (Evolution 300 UV-Vis Spectrophotometer, Thermo Fisher Scientific, Miami, OK, USA).

### 4.3. Preparation of the Ecklonia stolonifera Extract

*Ecklonia stolonifera* samples were collected from Jeju Island, Jeju Province, Korea. Botanical identification was made by Wook Jae Lee (Jeju Technopark, Jeju, Korea), and a sample specimen was deposited at the herbarium of the Jeju Biodiversity Research Institute, Jeju, Korea. The dried *Ecklonia stolonifera* samples were homogenized into a fine powder using an electric mixer (HMF-3100S, Hanil Electric, Seoul, Korea). The *Ecklonia stolonifera* extract was prepared by dissolving the powder in 80% ethanol at room temperature. This solution was then filtered and concentrated using a rotary vacuum evaporator (Buchi Rotavapor R-144, Buchi Labortechnik, Flawil, Switzerland).

### 4.4. Synthesis and Physicochemical Characterization Of ES-GNPS

The synthesis and physicochemical characterization of ES-GNPs were determined following the method described by [18]. In brief, to synthesize the ES-GNPs, an aqueous solution consisting of 1 mM gold (III) chloride solution (HAuCl_4_) was mixed with the ES extract (2 mg/mL). The mixture was rigorously stirred and incubated at 25 °C for 15 min. The color change from yellow to violet after 15 min indicated the formation of ES-GNPs. ES-GNPs were detected using an Evolution 300 UV-Vis spectrophotometer (Thermo Fisher Scientific, Miami, OK, USA) from 300 to 800 nm. The particle size, zeta-potential, and polydispersity index (PDI) of ES-GNPs were determined at 25 ℃ by DLS technique using Zetasizer Nano ZS90 (Malvern Instruments, Malvern, UK). The ES-GNPs were placed in a disposable zeta cell at 25 ℃. X-ray diffraction (XRD) was performed using an X’Pert3 Powder X-ray Diffractometer (Malvern Panalytical, Malvern, UK) operating at a scanning range of 30 to 80; voltage, 40 kV; and current, 30 mA. Fourier-transform infrared spectroscopy (FT-IR) was carried out using KBr pellets on a Perkin Elmer Spectrum GX FT-IR spectrophotometer operating in the range of 4000 and 400 cm^−1^. The surface morphology, crystallinity, and chemical composition of the ES-GNPs were examined using high resolution transmission electron microscopy (HR-TEM), selected area electron diffraction (SAED), fast Fourier transform (FFT), and high-angle annular dark field (HAADF) analysis. Energy dispersive spectroscopy (EDS) was performed on Thermo Scientific (FEI) Talos F200X G2 TEM.

### 4.5. Photocatalytic Activities of ES-GNPS

The photocatalytic activity of ES-GNPs was evaluated by observing the degradation of methylene blue and rhodamine B [30]. In brief, ES-GNPs were added to a solution containing methylene blue (0.8 mM) and rhodamine B (0.05 mM); then, ice cold sodium borohydride (0.06 M) solution was added. The degradation of the dye was monitored on an Evolution 300 UV-Vis spectrophotometer (Thermo Fisher Scientific, Miami, OK, USA) in the range of 300–800 nm at regular intervals (1 min).

### 4.6. Cell Culture and Establishment of a Cellular Model of UVA Irradiation-Induced Ssenescence

Human dermal fibroblasts were obtained from Lonza (Walkersville, MD, USA). They were maintained in Dulbecco’s Modified Eagle’s Medium (DMEM) (GIBCO, Grand Island, NY, USA), supplemented with 10% heat-inactivated fetal bovine serum (FBS) and 1% penicillin/streptomycin (Invitrogen, Carlsbad, USA), under a humidified atmosphere (95% air, 5% CO_2_) at 37 °C. All experiments were performed with HDFs from passage 4–6. First, HDFs were cultured in 6-well plates at a density of 3 × 10^4^ cells per well for 24 h. Afterward, HDFs were treated with ES-GNPs (100 µg/mL) for 24 h; then, HDFs were exposed to UVA irradiation (Bio-Link BLX-365; Villber-Lourmat, Eberhardzell, Germany) with 5 × 8 W tubes (365 nm) serving as the UVA source. HDFs were washed twice with phosphate-buffered saline (PBS) and were then irradiated with UVA at 10 J/cm^2^. Then, PBS was removed, and HDFs were retreated with ES-GNPs (100 µg/mL) for 24 h. The control comprised untreated and unirradiated cells.

### 4.7. Cell Counting Kit-8 Assay

The viability of HDFs was assessed using the CCK-8 assay (Sigma, USA) in accordance with the manufacturer’s instructions. The optical density value was determined using a FLUOstar Omega Plate Reader (BMG Labtech, Ortenberg, Germany) at 450 nm.

### 4.8. Senescence-Associated β-Galactosidase (SA-β-gal) Assay

SA-β-gal activity was determined using the senescence β-galactosidase staining kit (Cell Signaling Technology; Beverly, MA, USA), and a fluorogenic substrate based Quantitative Cellular Senescence Assay Kit (Cell Biolabs, Inc.; San Diego, CA, USA), according to the manufacturer’s instructions. The proportions of SA-β-gal staining in the HDFs were represented as a percentage of the total number of HDFs counted in the optical field. SA-β-gal-stained HDFs were identified based on the fluorescent intensity that was recorded on a Flow Cytometer (Beckman Coulter FC500, Pasadena, CA, USA).

### 4.9. Measurement of Intracellular ROS Production and Lysosome Content

Intracellular ROS levels and lysosome content in the treated HDFs were evaluated using the ROS assay kit (CM-H_2_DCFDA, Thermo Fisher Scientific, Inc., Miami, OK, USA) and Lysotracker Green DND-26 (Cell signaling Technology, Beverly, MA, USA), according to manufacturer’s instructions. In brief, after UVA irradiation, the HDFs were rinsed with PBS and incubated with CM-H_2_DCFDA or Lysotracker Green DND-26 for 30 min in the dark. The fluorescent intensity was proportional to the intracellular ROS levels and lysosome content. Thereafter, intracellular ROS levels were determined based on the fluorescent intensity that was recorded on a Flow Cytometer (Beckman Coulter FC500, Pasadena, CA, USA).

### 4.10. Cell Cycle Assay

Flow cytometric analysis was conducted to investigate the cell cycle distribution of the UVA-irradiated HDFs. Briefly, HDFs were collected by trypsinization and washed three times with PBS. Subsequently, HDFs were stained with Propidium Iodide ReadyProbes Reagent (Thermo Fisher Scientific, Milpitas, CA, USA), according to the manufacturer’s protocol. Finally, cell cycle analysis was performed on a Flow Cytometer Cytomics FC 500 (Beckman Coulter, Pasadena, CA, USA).

### 4.11. Total RNA Extraction and Quantitative Real Time PCR Analysis

The Total RNA Extraction and Quantitative Real Time PCR was used with some alterations to determine the mRNA levels [18]. Total RNA was isolated from HDFs using RNeasy Mini kit (QIAGEN, Hilden, Germany), and cDNA was synthesized by reverse transcription using high-capacity cDNA reverse transcription kit (Thermo Fisher Scientific, Miami, OK, USA). Quantitative Real Time PCR (qRT-PCR) was performed using SYBR Green qPCR master mixes (Thermo Fisher Scientific, Miami, OK, USA). Real time PCR assays were performed according to the manufacturer’s instructions. The relative amount of target mRNA was determined using the Ct method by normalizing target mRNA Ct values to those for GAPDH (ΔCt). The real-time PCR cycling conditions were 95 °C for 5 min, 40 cycles for 30 s at 95 °C, 30 s at 55 °C, and 30 s at 72 °C, followed by fluorescence measurement. The primer sequences used were as follows: MMP-1-sense (5′- tctgacgttgatcccagagagcag-3′), MMP-1-anti-sense (5′- cagggtgacaccagtgactgcac-3′), MMP-3-sense (5′-gagagcagaagaccgaaagga-3′), MMP-3-anti-sense (5′- cacaacaccacgttatcggg-3′), GAPDH-sense (5′-aggtggtctcctctgacttc-3′), and GAPDH-anti-sense (5′-taccaggaaatgagcttgac-3′).

### 4.12. MMP-1 and MMP-3 Flow Cytometry

Antibodies against MMP-1 (IC9011P) and MMP-3 (IC513P) were procured from R & D Systems Technology, Inc. (Beverly, MA, USA). The expression of MMP-1 and MMP-3 proteins was determined by means of flow cytometry using the PE-conjugated anit-MMP-1 and MMP-3. Isotype control is Mouse IgG1 PE-conjugated Antibody (IC002P, 5 µL/10^5^ cells). Briefly, HDFs were fixed and permeabilized using the FIX & PERM Cell Permeabilization Kit (Thermo Fisher Scientific, Miami, OK, USA). Thereafter, HDFs were incubated with the anti-MMP-1 (5 µL/10^5^ cells) and MMP-3 antibody (5 µL/10^5^ cells). Finally, the HDFs were analyzed for MMP-1 and MMP-3 protein expression using the Flow Cytometer Cytomics FC 500 (Beckman Coulter, Pasadena, CA, USA).

### 4.13. Western Blotting Analysis

HDFs were harvested and split with M-PER Mammalian Protein Extraction Reagent (Thermo Fisher Scientific, Miami, OK, USA) according to manufacturer’s instructions. The protein concentration was determined using Bio-Rad protein assay kits (Bio-rad, Hercules, CA, USA). Cell lysates were separated using 7–12% SDS-PAGE. Proteins were then electrotransferred onto a PVDF membrane (Amersham Biosciences, Piscataway, NJ, USA). After blocking with 5% BSA for 1 h at room temperature, the PVDF membranes were incubated with the following primary antibodies: Anti-p16 (1:500, #92803), anti-p21 (1:500, #2947), anti-p-p53 (1:500, #9286), anti-p53 (1:500, #2524) and anti-β-tubulin (1:1000, #2144) (Cell Signaling, Waltham, MA, USA). After washing three times for 10 min with Tris-buffered saline and Tween 20 (TBST), the membranes were incubated with horseradish peroxidase-conjugated secondary antibodies (antirabbit IgG (1:1000, #7074), antimouse IgG (1:1000, #7076), Cell Signaling, Waltham, MA, USA) for 1 h at room temperature. The protein bands were detected using an enhanced Pierce ECL Western Blotting Substrate (Thermo Fisher Scientific, Miami, OK, USA) and quantified as the ratio of the intensity of the band to the intensity of the α-tubulin band. Quantification was performed using an ImageQuant 350 analyzer (Amersham Biosciences, Piscataway, NJ, USA).

### 4.14. Enzyme-Linked Immunosorbent Assay (ELISA)

To examine whether ES-GNPs affected the secretion of MMP-1 and MMP-3, an ELISA was performed. The levels of MMP-1 (# EHMMP1) and MMP-3 (# BMS2014-3) in the supernatant were detected by commercial ELISA kits (Thermo Fisher Scientific, Miami, OK, USA) according to the manufacturer’s recommendations, followed by absorbance detection at 450 nm using a full wavelength microplate reader.

### 4.15. Statistical Analysis

All data are represented as means ± SEM of three independent replicates for each group. Comparisons were conducted using the Statistical Package for the Social Sciences software, version 17.0 (SPSS Inc. Released 2008. SPSS Statistics for Windows, Version 17.0. Chicago: SPSS Inc). Student’s *t*-test and one-way analysis of variance (ANOVA) were used to evaluate the differences among groups. A *p*-value of <0.01 or <0.05 was considered to indicate a statistically significant difference.

## 5. Conclusions

The *Ecklonia stolonifera* extract can serve as a metal nanoparticle reducing agent due to the presence of many phlorotannins, phenolic compounds, terpenoids, steroids, and vitamins that can be used during the reduction, capsulation, and stabilization of ES-GNPs. This fact was confirmed in the case of ES-GNPs by UV-vis spectra, DLS, HR-TEM, EDS, XRD, and FT-IR. ES-GNPs. ES-GNPs exhibit effective photocatalysis by degrading methylene blue and rhodamine B. Our study demonstrated that ES-GNPs protected HDFs from UVA irradiation-induced cellular senescence by inhibiting the SA-β-galactosidase activity, reducing intracellular ROS production, lysosome content, and inhibiting G1 arrest and senescence related proteins. Additionally, these anticellular senescence effects can be mediated via the inhibition of MMP-1/-3 expression and secretion. Considering the biological functions of ES-GNPs, we speculate that ES-GNPs could serve as potential candidates for the treatment of skin-photoaging.

## Figures and Tables

**Figure 1 marinedrugs-18-00433-f001:**
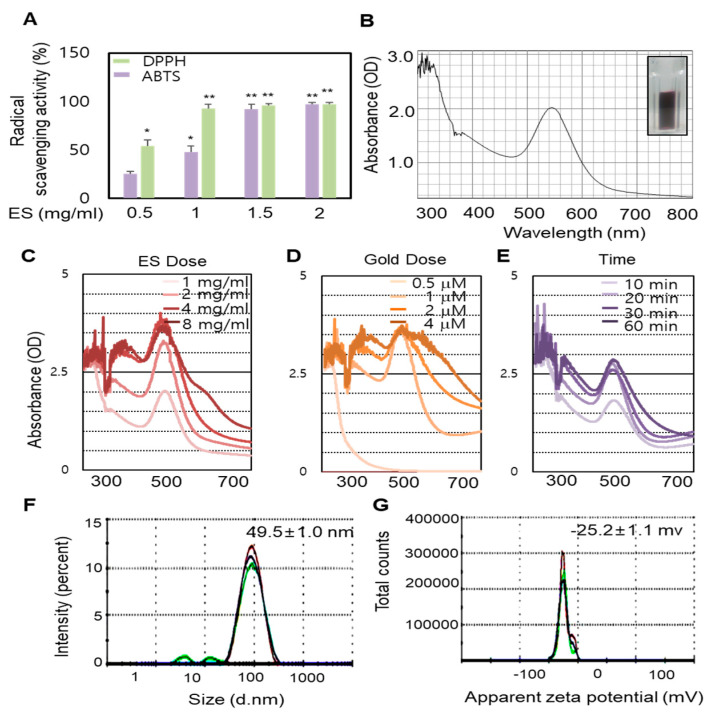
Radical scavenging activity of *Ecklonia stolonifera* (ES) extract and optimization of *Ecklonia stolonifera* (ES)-GNPs (gold nanoparticles). (**A**) 2,2′-Azino-bis(3-ethylbenzothiazoline-6-sulphonic acid) (ABTS) and 2,2-diphenyl-1-picrylhydrazyl (DPPH) radical scavenging activity of ES extract. (**B**) UV-visible spectrum analysis of ES-GNPs. Optimization of ES-GNP synthesis using ES extract dose (**C**), gold (III) chloride dose (**D**), and synthetic times (**E**). The determination of hydrodynamic size (**F**) and zeta potential (**G**) of ES-GNPs from DLS (dynamic light scattering). The results are given as the UV-visible spectrum, average particle size, zeta-potential, and polydispersity index (PDI) acquired from the examination of three different batches, each of them measured three times. Three independent experiments are expressed as mean ± standard error of the mean (SEM). * *p* < 0.05 and ** *p* < 0.01 compared to the UVA-irradiated group.

**Figure 2 marinedrugs-18-00433-f002:**
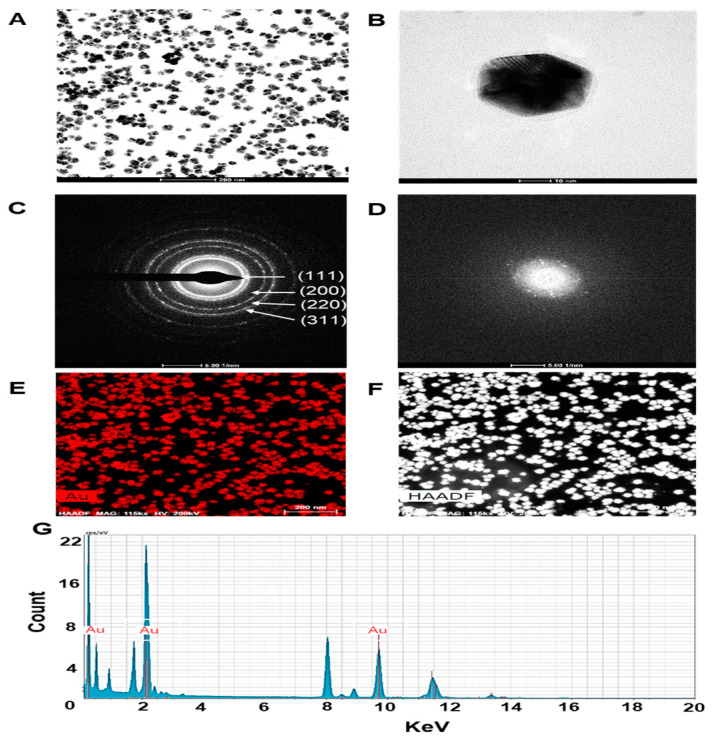
Characterization of ES-GNPs. High resolution transmission electron microscopy (HR-TEM) images at (**A**) low magnification, (**B**) high magnification, (**C**) Selected area electron diffraction (SAED) pattern, (**D**) Fast Fourier transform (FFT) pattern (**E**,**F**) High-angle annular dark field (HAADF) image, and (**G**) Energy dispersive spectroscopy (EDS) analysis of ES-GNPs. The results are given as the HR-TEM images acquired from the examination, each of them measured three times.

**Figure 3 marinedrugs-18-00433-f003:**
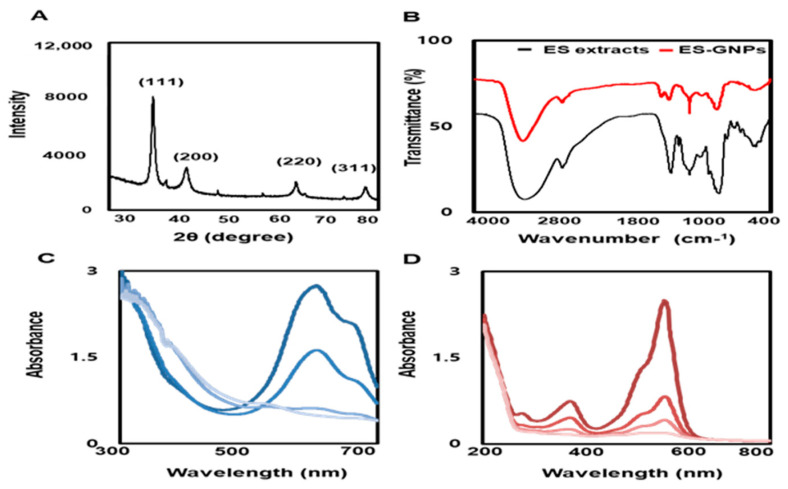
Physicochemical characterization of ES-GNPs. (**A**) X-ray diffraction (XRD) pattern and (**B**) Fourier-transform infrared spectroscopy (FT-IR) spectra of ES-GNPs, and UV-visible spectrum of (**C**) methylene blue and (**D**) Rhodamine B after addition of ES-GNPs. The results are given as the XRD pattern, FTIR spectra, UV-visible spectrum acquired from the examination, each of them measured three times.

**Figure 4 marinedrugs-18-00433-f004:**
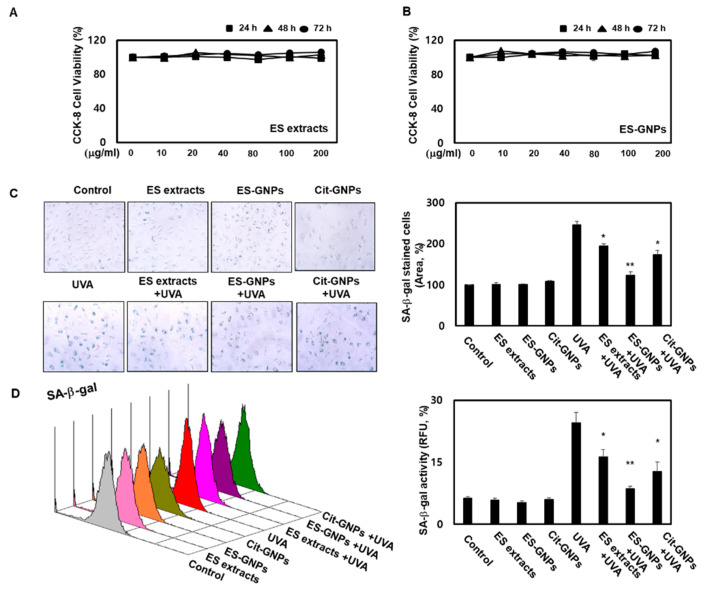
ES-GNPs attenuate the SA-β-galactosidase activity in UVA-irradiated human dermal fibroblasts. (**A**,**B**) Cell viability of ES extracts and ES-GNPs for 24 h, 48 h and 72 h, respectively, were detected by the CCK-8 (cell count kit 8) assay. (**C**) SA-β-galactosidase activity was detected by using the Senescence β-Galactosidase Staining Kit. (**D**) Fluorometric SA-β-galactosidase activity was detected by using the Quantitative Cellular Senescence Assay Kit and expressed as relative fluorescence unit (RFU). Three independent experiments are expressed as mean ± standard error of the mean (SEM). * *p* < 0.05 and ** *p* < 0.01 compared to the UVA-irradiated group. Control: untreated and unirradiated cells. ES extracts: ES extracts treated cells. ES-GNPs: ES-GNPs treated cells. Cit-GNPs: Citric acid (0.25 mM)-GNPs treated cells. UVA: UVA irradiated cells. ES extracts + UVA: ES extracts treated and then UVA irradiated cells. ES-GNPs + UVA: ES-GNPs treated and then UVA irradiated cells. Cit-GNPs + UVA: Citric acid-GNPs treated and then UVA irradiated cells.

**Figure 5 marinedrugs-18-00433-f005:**
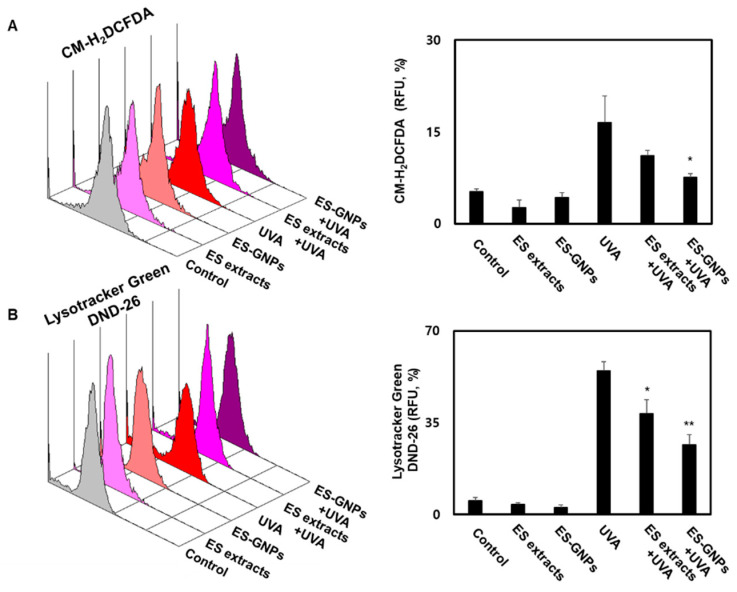
ES-GNPs inhibited the ROS production and lysosome content in UVA-irradiated human dermal fibroblasts. (**A**) The ROS generation was measured by flow cytometry using CM-H2DCFDA General Oxidative Stress Indicator and expressed as relative fluorescence unit (RFU). (**B**) The lysosome content was measured by Flow cytometry using LysoTracker Green DND-26 and expressed as relative fluorescence unit (RFU). Three independent experiments are expressed as mean ± SEM. * *p* < 0.05 and ** *p* < 0.01 compared to the UVA group. Control: untreated and unirradiated cells. ES extracts: ES extracts treated cells. ES-GNPs: ES-GNPs treated cells. UVA: UVA irradiated cells. ES extracts + UVA: ES extracts treated and then UVA irradiated cells. ES-GNPs + UVA: ES-GNPs treated and then UVA irradiated cells.

**Figure 6 marinedrugs-18-00433-f006:**
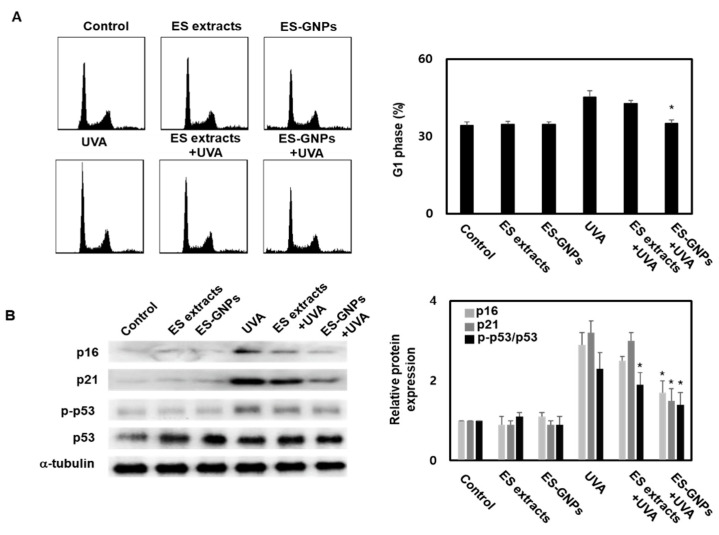
ES-GNPs inhibited the G1 arrest and protein expression of p16, p21 and p-p53 in UVA-irradiated human dermal fibroblasts. (**A**) The G1 arrest was measured by Flow cytometry using a cell cycle kit. (**B**) The protein expression of p16, p21, p-p53, p53 and β-tubulin were measured by western blot. Three independent experiments are expressed as mean ± SEM. * *p* < 0.05 compared to the UVA group. Control: untreated and unirradiated cells. ES extracts: ES extracts treated cells. ES-GNPs: ES-GNPs treated cells. UVA: UVA irradiated cells. ES extracts + UVA: ES extracts treated and then UVA irradiated cells. ES-GNPs + UVA: ES-GNPs treated and then UVA irradiated cells.

**Figure 7 marinedrugs-18-00433-f007:**
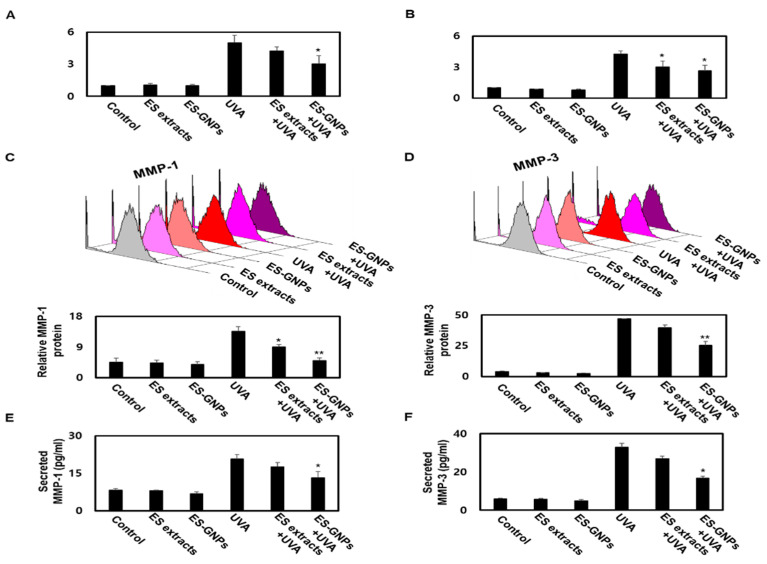
ES-GNPs downregulate the transcriptional, translational expression and secretion of MMP-1 and MMP-3 in UVA-irradiated human dermal fibroblasts. Real time PCR was carried out to evaluate the mRNA expression of MMP-1 (**A**) and MMP-3 (**B**). Flow cytometry was carried out to evaluate the protein expression of MMP-1 (**C**) and MMP-3 (**D**). The content of MMP-1 (**E**) and MMP-3 (**F**) in cell culture supernatants was measured using ELISA kit. Three independent experiments are expressed as mean ± SEM. * *p* < 0.05 and ** *p* < 0.01 compared to the UVA-irradiated group. Control: untreated and unirradiated cells. ES extracts: ES extracts treated cells. ES-GNPs: ES-GNPs treated cells. UVA: UVA irradiated cells. ES extracts + UVA: ES extracts treated and then UVA irradiated cells. ES-GNPs + UVA: ES-GNPs treated and then UVA irradiated cells.

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
