# Peer review of "Gold Nanoparticles Using Ecklonia stolonifera Protect Human Dermal Fibroblasts from UVA-Induced Senescence through Inhibiting MMP-1 and MMP-3"

_marinedrugs, 2020, doi:10.3390/md18090433_

Round 1
Reviewer 1 Report
The paper by Jun and coworkers is quite interesting. It seems a fairly well-conducted study of the subject and I think it deserves publication.
There are minor language imperfections hat authors may easily correct after careful reading. Authors should also put more attention in the comments to the figures that should report some more elements concerning the meaning of the different panels. Another point I would like to raise is the necessity of an abbreviation table that would help the reader.
Besides these minor issues there is a major one the authors should solve for the publication.
The experiments illustrated in Figures 5, 6, 7 should include also an additional control, i.e. citrate-coated nanoparticles that were included only in the experiment illustrated in Figure 4, namely the SA-b-galactosidase activity assay. In addition to the (possible) better performance of the ES-GNPs in the experiments of Figures 5, 6 and 7, other motivations that justify the use of ES-GNPs instead of the citrate coated gold nanoparticles may be the ease of control of the reduction of Au(III) and the control on the size of gold nanoparticles.
With those amendments the paper should become suitable for publication.
Author Response
Thank you for reviewing and providing your comments on my manuscript.
I have revised my manuscript in accordance with the comments from the Reviewer, and my point-by-point responses are listed below.
The paper by Jun and coworkers is quite interesting. It seems a fairly well-conducted study of the subject and I think it deserves publication.
There are minor language imperfections hat authors may easily correct after careful reading.
=> Thank you for your comment. In accordance with the instructions, the manuscript has been edited and revised by a professional English language-editing service (PUSU-5179).
Authors should also put more attention in the comments to the figures that should report some more elements concerning the meaning of the different panels. Another point I would like to raise is the necessity of an abbreviation table that would help the reader.
=> In accordance with the Reviewer’s comment, we have included the abbreviation. “Abbreviations defined in parentheses the first time they appear in the abstract, main text, and in figure or table captions and used consistently thereafter.”
Besides these minor issues there is a major one the authors should solve for the publication.
The experiments illustrated in Figures 5, 6, 7 should include also an additional control, i.e. citrate-coated nanoparticles that were included only in the experiment illustrated in Figure 4, namely the SA-b-galactosidase activity assay. In addition to the (possible) better performance of the ES-GNPs in the experiments of Figures 5, 6 and 7, other motivations that justify the use of ES-GNPs instead of the citrate coated gold nanoparticles may be the ease of control of the reduction of Au(III) and the control on the size of gold nanoparticles.
With those amendments the paper should become suitable for publication.
=> We also studied the effects of the citric acid capped gold nanoparticles on the SA-β-galactosidase activity assay in UVA-irradiated HDFs (Figure 4). citric acid capped gold nanoparticles also inhibit the SA-β-galactosidase activity assay in UVA-irradiated HDFs. Due to the properties, we will further investigate the relationship with ROS, apoptosis related protein, MMP-1and MMP-3 expression and characteristic analysis of senescence-related signaling in HDFs.
In the present study, GNPs were synthesized using an eco-friendly protocol using Ecklonia stolonifera for as reducing, stabilizing, and capping agents for medical purposes. The successful synthesis of ES-GNPs was confirmed by UV-vis spectra, DLS, HR-TEM, EDS, XRD and FT-IR. To the best of our knowledge, this is the first report indicating that ES-GNPs exhibit anti-senescence properties in UVA-irradiated human dermal fibroblast.
Reviewer 2 Report
The authors describe the work entitled "Facile Gold Nanoparticles Using Ecklonia Stolonifera Protects Human Dermal Fibroblasts From UVA-3 Induced Senescence Through Inhibiting MMP-1 and MMP-3". These results findings indicated that these facile and optimum ES-GNPs exerted an anti-senescence effect on UVA-36 irradiated HDFs via inhibiting MMP-1/-3 expression. In addition, ES-GNPs can be potentially used to treat photoaging of the skin. The manuscript is nicely arranged and should be accepted after following minor revision:
(1) Authors should include "Conclusion" section
Author Response
Thank you for reviewing and providing your comments on my manuscript.
I have revised my manuscript in accordance with the comments from the Reviewer, and my point-by-point responses are listed below.
The authors describe the work entitled "Facile Gold Nanoparticles Using Ecklonia Stolonifera Protects Human Dermal Fibroblasts From UVA-3 Induced Senescence Through Inhibiting MMP-1 and MMP-3". These results findings indicated that these facile and optimum ES-GNPs exerted an anti-senescence effect on UVA-36 irradiated HDFs via inhibiting MMP-1/-3 expression. In addition, ES-GNPs can be potentially used to treat photoaging of the skin. The manuscript is nicely arranged and should be accepted after following minor revision:
(1) Authors should include "Conclusion" section
=> In accordance with the Reviewer’s comment, we have included the Conclusion. (line 464).
Reviewer 3 Report
In the paper entitled “Facile Gold Nanoparticles Using Ecklonia Stolonifera Protects Human Dermal Fibroblast From UVA-Induced Senescence Through Inhibiting MMP-1 and MMP-3” by Jun ES et al, they showed that Ecklonia stolonifera gold nanoparticles (ES-GNPs) can significantly inhibit UVA-induced ROS generation, MMP-1/3 expression and G1 arrest. The authors presented results clearly and conclusions are hardly controversial. I think this paper is suitable for publication in Marine Drugs.
Author Response
Thank you for reviewing and providing your comments on my manuscript.
I have revised my manuscript in accordance with the comments from the Reviewer, and my point-by-point responses are listed below.
In the paper entitled “Facile Gold Nanoparticles Using Ecklonia Stolonifera Protects Human Dermal Fibroblast From UVA-Induced Senescence Through Inhibiting MMP-1 and MMP-3” by Jun ES et al, they showed that Ecklonia stolonifera gold nanoparticles (ES-GNPs) can significantly inhibit UVA-induced ROS generation, MMP-1/3 expression and G1 arrest. The authors presented results clearly and conclusions are hardly controversial. I think this paper is suitable for publication in Marine Drugs.
=> Thank you for reviewing and providing your comments on my manuscript.